# MULTI-SESSION CLIENT-CENTERED TREATMENT OUTCOME EVALUATION IN PSYCHOTHERAPY

## ABSTRACT

In psychotherapy, therapeutic outcome assessment, or treatment outcome evaluation, is essential for enhancing mental health care by systematically evaluating therapeutic processes and outcomes. Existing large language model approaches often focus on therapist-centered, single-session evaluations, neglecting the client's subjective experience and longitudinal progress across multiple sessions. To address these limitations, we propose IPAEval, a client-Informed Psychological Assessment-based Evaluation framework that automates treatment outcome evaluations from the client's perspective using clinical interviews. IPAEval integrates cross-session client-contextual assessment and session-focused client-dynamics assessment to provide a comprehensive understanding of therapeutic progress. Experiments on our newly developed TheraPhase dataset demonstrate that IPAEval effectively tracks symptom severity and treatment outcomes over multiple sessions, outperforming previous single-session models and validating the benefits of items-aware reasoning mechanisms.

## 1 INTRODUCTION

In psychotherapy, therapeutic outcome assessment, a.k.a treatment outcome evaluation under clinical settings, refers to the systematic evaluation of therapeutic processes and outcomes (Groth-Marnat, 2009), focusing on factors such as therapist effectiveness (Johns et al., 2019) and treatment efficacy (Jensen-Doss et al., 2018) to improve mental health care delivery. It plays a significant role in enhancing the quality and effectiveness of mental health care by providing actionable insights that guide therapists in refining their treatment approaches (Wampold & Imel, 2015), ultimately leading to better client outcomes and improved therapeutic relationships in real-world clinical practice (Maruish & Leahy, 2000).

Over the last couple of years, the emergence of large language models has demonstrated their effectiveness in automatic evaluations, showing a high degree of alignment with human judgment when provided with proper instruction and contextual guidance (Liu et al., 2023; Li et al., 2024b; Kim et al., 2024). This aligns with the "LLMs-as-a-judge" paradigm, where LLMs are employed to simulate human evaluators by providing assessments based on natural language input (Zheng et al., 2023; Wang et al., 2024b). This paradigm has been extended to therapeutic outcome assessment by harnessing LLMs' ability to model complex therapeutic procedures and interactions, offering a novel pathway for automating the assessment of therapeutic efficacy (Chiu et al., 2024; Lee et al., 2024; Li et al., 2024a).

In the assessment, compared to psychometric tests (Furr, 2020) that are often constrained by the limitations of self-reported data, susceptibility to social desirability biases (Braun et al., 2001; Paulhus, 2017), clinical interviews not only provide richer, more nuanced insights into the client's emotional and behavioral states but also offer data that is more readily obtainable through natural, conversational interactions. Therefore, many recent works leverage clinical interviews, potentially enriched by the client's profile (Lee et al., 2024), to evaluate therapists from multiple perspectives, including behavioral labels (Chiu et al., 2024), skills adherence (Lee et al., 2024), and therapeutic rapport (Li et al., 2024a; Yosef et al., 2024), offering a holistic view of their effectiveness in psychotherapy.

While the above therapist-centered assessments focus on evaluating the therapist's techniques and adherence to therapeutic models, they often overlook the subjective experience and evolving needs of the client, limiting the depth of the evaluation (Wang et al., 2024a; Yosef et al., 2024). In contrast,

| Method | Perspective | Theory Adherence | Reasoning | Evaluation Target |
|---|---|---|---|---|
| CPsyCoun (Zhang et al., 2024) | Therapist | ✗ | ✗ | Single Session |
| Cactus (Lee et al., 2024) | Therapist | ✓ | ✗ | Single Session |
| ClientCAST (Wang et al., 2024a) | Client | ✓ | ✗ | Single Session |
| IPAEval (Ours) | Client | ✓ | ✓ | Multiple Sessions |

Table 1: A comparison of IPAEval with other treatment outcome evaluation methods. **Perspective** indicates whether the evaluation is conducted from the therapist's or the client's point of view. **Theory Adherence** signifies whether the method is grounded in established psychological theories. **Reasoning** denotes whether the method involves generating intermediate reasoning steps before arriving at the final evaluation results. **Evaluation Target** refers to whether the method evaluates a single session or multiple sessions.

client-centered assessments, such as *treatment outcome evaluation* in common practice, prioritize the client's perspective, offering a more comprehensive understanding of therapy's impact by capturing changes in the client's emotional, cognitive, and behavioral states across sessions (Hatfield & Ogles, 2004; Rogers, 2012). Although a concurrent work, ClientCAST (Wang et al., 2024a), presents an LLM-based client simulator for treatment outcome evaluations, which focuses on reducing harmful outputs and improving answering consistency, we stand fundamentally apart and never fabricate client responses that could distort the evaluation of treatment outcomes. What's worse, almost all previous approaches focus on evaluating individual therapy sessions in isolation, without considering the broader context of the client's journey across multiple sessions. This narrow scope limits the ability to assess longitudinal progress or capture the dynamic shifts in a client's mental state and therapeutic needs over time, which are crucial for a comprehensive treatment outcome evaluation (Hayes & Andrews, 2020).

Motivated by the above therapist-centered and single-session limitations (please see Table 1 for comparisons), we design a new evaluation framework, dubbed client-**I**nformed **P**sychological **A**ssessment-based **Eval**uation (IPAEval), for treatment outcomes in the format of clinical interviews.

Specifically, to achieve treatment outcome evaluation, we formulate an information extraction task that leverages clinical interviews to automatically populate psychometric tests for psychological assessments, bridging the gap between subjective client dialogues and standardized metrics. As such, treatment outcomes are evaluated through these assessments of clients conducted both before and after therapy, allowing for a more comprehensive understanding of therapeutic progress. Upon this new framework, we first propose a cross-session client-contextual assessment module that integrates client history and contextual information across multiple sessions to enhance the accuracy of psychological assessments. Then, we present a session-focused client-dynamics assessment module that evaluates the effectiveness of individual therapy sessions by tracking real-time client responses and treatment outcomes within each session. In the meantime, to boost reasoning capability in the extraction, we also present an items-aware reasoning prompt technique for psychometric test-oriented rationale generation.

To evaluate the proposed framework, we first develop a new dataset, called TheraPhase, based on CPsyCoun (Zhang et al., 2024), which includes transcripts from initial and final therapy sessions. This dataset offers valuable insights into therapy progress and serves as a key resource for evaluating psychological assessments and treatment outcomes across multiple sessions. Then, we tested nine LLMs, including closed-source models. These models were evaluated for their performance in psychological assessments and treatment outcome prediction, particularly in multi-session evaluations. IPAEval consistently tracked symptom severity and treatment outcomes across multiple sessions, a capability lacking in previous single-session models. Our ablation study confirmed that the items-aware reasoning mechanism significantly boosts model performance in both symptom detection and outcome prediction.

## 2 RELATED WORK

**Therapist Assessment using LLMs.** LLMs' role-playing capabilities have led to increased interest in developing Role-Play Therapists (Chen et al., 2023; Chiu et al., 2024; Lee et al., 2024), but the lack of automated metrics for evaluating t herapist is a significant challenge. CPsyCoun (Zhang et al., 2024) employs an LLM-based evaluation method from the therapist's perspective to assess single session, specifically evaluating the therapist's comprehensiveness, professionalism, authenticity, and safety.

Lee et al. (2024), Li et al. (2024a), and Yosef et al. (2024) similarly adopt a therapist's perspective with LLM-based evaluation, but they address CPsyCoun's lack of support from psychological theories by employing the Cognitive Therapy Rating Scale (Goldberg et al., 2020) for CBT skills assessment and the Working Alliance Inventory (Hatcher & Gillaspy, 2006) for evaluating the therapeutic relationship. Notably, BOLT (Chiu et al., 2024) applied LLMs to identify therapist behaviors, evaluating the quality of dialogue sessions based on the frequency and sequence of LLM therapist behaviors. Clinical evidence (Goodson et al., 2017; Mason et al., 2016) shows that better therapists are linked to improved outcomes, but evaluating therapists alone may miss how much the client is benefiting (Robinson, 2009). The treatment outcome evaluation based on client-centered psychological assessment focuses more on results, specifically determining whether the therapy has brought about meaningful changes in the client's life, which is the ultimate goal of the treatment (Groth-Marnat, 2009).

**Client-centered Psychological Assessment.** Client-centered psychological assessment combines psychometric tests and clinical interviews to provide a comprehensive under-standing of the individual (Spoto et al., 2013). Psychometric tests offer standard-ized data on psychological traits, while clinical interviews give deeper insights into the client's personal experiences (Groth-Marnat, 2009). While tests may overlook certain nuances, interviews address these gaps by exploring context and individual differences. In clinical practice, the use of multiple assessment methods ensures a more complete understanding of the client (Meyer et al., 2001; Groth-Marnat, 2009). Leveraging the powerful general language processing capabilities (Luo et al., 2023; Zhao et al., 2023b) of LLMs enables the realization of com-plex and diverse assessment tasks. This contrasts with earlier approaches that focused solely on detecting individual psy-chological symptoms (Ji et al., 2022; Zhai et al., 2024), and a substantial body of research (Galatzer-Levy et al., 2023; Arcan et al., 2024; Rosenman et al., 2024) supports this ad-vancement. For instance, several studies have utilized LLMs to analyze interviews (Gratch et al., 2014), assessing depres-sion and Post-Traumatic Stress Disorder scores based on widely used psychometric tests like (Kroenke et al., 2009) and PCL-C (Weathers et al., 1994). However, precise psy-chological assessments enable therapists to grasp the client's psychological state, but a psychological assessment alone cannot determine whether the treatment has brought about positive changes for the client.

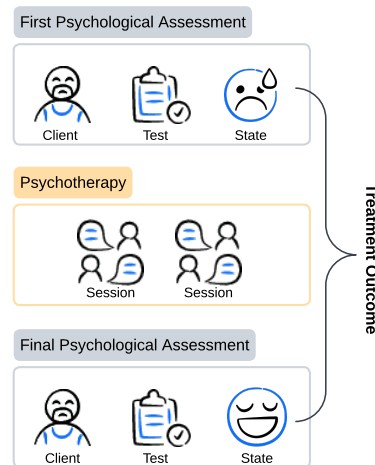

Figure 1: What is Treatment Outcome?

Treatment outcomes evaluation complements psychological assessment by measuring the effective-ness of interventions over time (Maruish & Leahy, 2000). While psychological assessments provide a snapshot of the client's mental state, as shown in the Figure 1, treatment outcomes evaluation focuses on tracking changes in symptoms and overall well-being throughout the therapeutic process. This dynamic evaluation allows therapists to determine whether the treatment has been successful and adjust strategies as needed to improve results.

## 3 METHODOLOGY

In this section, starting with a formal task definition (§3.1), we elaborate on our evaluation frame-work, called client-Informed Psychological Assessment-based Evaluation (IPAEval), which is mainly composed of 1) *cross-session client-contextual assessment* module (§3.2) for client-tracking psy-chological assessment and a *session-focused client-dynamics assessment* module (§3.3) to derive session-informed treatment outcome evaluation. Please see Figure 2 for an overall illustration of our framework. As there is no precursor in clinical interviews-based treatment outcome evaluation, we, therefore, curate a new dataset, called TheraPhase, as a testbed for our proposed IPAEval framework.

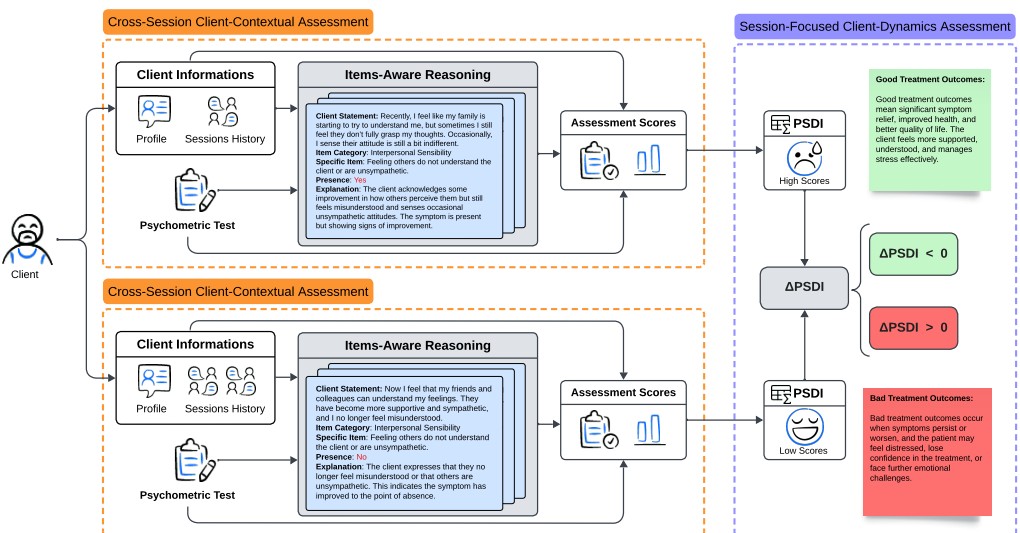

Figure 2: An illustration of client-informed psychological assessment-based evaluation (IPAEval).

## 3.1 TASK DEFINITION OF IPAEVAL FRAMEWORK

To deliver treatment outcome evaluations for a certain client with profile $p$, based on multiple sessions $[s_1, s_2, \dots]$, we aim to evaluate the efficacy of a certain session $s_k$ in clinical interviews as treatment outcome. Without sacrificing generalization to one whole treatment composed of several sessions, $s_k$ here could be a combination of the sessions. To achieve the above, we need to split the task into two sequential sub-tasks – psychological assessment ($\boldsymbol{a}_k$) based on client information and treatment outcome evaluation ($\boldsymbol{e}_k$) based on two or more assessments. To derive $\boldsymbol{a}_k$ and $\boldsymbol{e}_k$, we first define client-informed input of an assessment after $k$-th session, i.e.,

$$c_k = p \oplus s_k \oplus h_k, \text{ where } h_k = [s_1, \dots, s_{i-1}] \oplus [\boldsymbol{a}_1, \dots] \oplus [\boldsymbol{e}_1, \dots]. \tag{1}$$

Here, $h_k$ denotes a set of meta client-contextual information from the past, e.g., past interviews $[s_1, \dots, s_{i-1}]$, past assessments $[\boldsymbol{a}_1, \dots]$, or/and past outcome evaluations $[\boldsymbol{e}_1, \dots]$. Upon this, we could easily define psychological assessment after $i$-th session as

$$\boldsymbol{a}_k = \mathrm{M}^{(\mathrm{a})}(c_k, \mathbb{T}), \tag{2}$$

where $\mathbb{T}$ is a set of psychometric tests as the criteria to evaluate the client information, and $\mathrm{M}$ denotes an approach to derive $\boldsymbol{a}_k$. Then, the treatment outcome evaluation for $k$-th session would defined as

$$\boldsymbol{e}_k = \mathrm{M}^{(\mathrm{e})}(c_k, \boldsymbol{a}_k, h_k), \tag{3}$$

In the remaining, we omit the index of session $k$ if no confusion is raised for clear annotations.

To handle the above two sequential sub-tasks, we detail our two modules, which aim to tackle the sub-tasks respectively, in the following.

## 3.2 CROSS-SESSION CLIENT-CONTEXTUAL ASSESSMENT

Existing research using client information with LLMs for mental health assessment, particularly for depression and PTSD, shows promising results (Galatzer-Levy et al., 2023; Arcan et al., 2024; Rosenman et al., 2024). However, these studies typically focus on specific symptoms and lack broad coverage of psychological conditions and transparency in interpreting scale results, which may erode trust among clinicians and clients, limiting clinical applications (Martin & Rouas, 2024).

To address these gaps, we introduce a two-stage prompt scheme that populates information from clinical interviews to fill psychometric tests by making the best of LLMs' capability in natural language understanding (Zhao et al., 2023a; Hua et al., 2024). It's applicable to various psychometric tests, specifically designed to provide interpretable psychological assessments. Without sacrificing generality, in this work we utilize the Symptom Checklist-90 (SCL-90) (Derogatis & Unger, 2010), a widely used and comprehensive psychometric test for screening psychological symptoms.

**Items-Aware Reasoning.** This stage aims to generate detailed reasoning for psychometric test items using LLMs, leveraging client information. Here, the items of the SCL-90 represent psychological symptoms. It correlates specific client information with the corresponding symptoms and items from the SCL-90, determining their presence and providing an interpretation. Given an out-of-the-box LLM (Dubey et al., 2024; Yang et al., 2024; Jiang et al., 2024) able to follow instructions, we first curate a prompt to steer the LLM to extract information from interviews to structured psychometric tests with thought augmentations. Inspired by a recent work Schulhoff et al. (2024), our prompt design integrates several components: a psychologist role, denoted by $r^{(pi)}$, skilled at recognizing symptoms, the SCL-90 as additional information $\mathbb{T}$, output formatting $o^{(pi)}$, and specific directives $d^{(pi)}$. Based on these, we can define psychometric interpretation prompt $p^{(pi)}$ as follows:

$$p^{(pi)} = f(r^{(pi)}, \mathbb{T}, o^{(pi)}, d^{(pi)}) \tag{4}$$

Furthermore, given the client information $c$, an LLM is prompted to generate items-aware reasoning results $\hat{\mathbb{X}}$:

$$\hat{\mathbb{X}} = \underset{\mathbb{X}}{\arg\max}\, P_{\text{LLM}}(\mathbb{X}|c, p^{(pi)}), \tag{5}$$

where $p^{(pi)}\, \hat{\mathbb{X}}$ represents a set of predicted items-aware reasoning results, each element in the set consisting of extracted client information, symptom category, specific symptom, presence, and a detailed explanation. It is noteworthy that this approach helps clinicians quickly trace the source of evidence for assessments and offers a clear pathway to understanding the interconnections and relevance of various symptoms presented by the client. The detailed prompt and an example of Items-Aware Reasoning are provided in Appendix A and Appendix C, respectively.

**Psychological Assessment.** The other stage is designed to harness the capabilities of LLMs in conducting psychological assessments based on client information and item-aware reasoning results. Similar to the items-aware reasoning stage, it comprises four main components: a psychologist role denoted by $r^{(sa)}$, skilled at symptom assessment, the SCL-90 psychometric test $\mathbb{T}$, alongside score criteria $s$ serving as additional information, output formatting implemented $o^{(sa)}$, and specific directives $d^{(sa)}$. Considering the practical constraints of client information, where not all 90 questions from the SCL-90 are likely to be addressed, we have simplified the scoring criteria. Instead of scoring each of the 90 items individually, the assessment has been adapted to score across 10 symptom dimensions derived from these items. Based on these, we can define symptom assessment prompt $p^{(sa)}$ as follows:

$$p^{(sa)} = f(r^{(sa)}, \mathbb{T}, s, o^{(sa)}, d^{(sa)}) \tag{6}$$

Formally, given the client information $c$ and items-aware reasoning result $\hat{\mathbb{X}}$ generated by LLM, an LLM is prompted to generate assessment scores $\hat{\boldsymbol{a}}$:

$$\hat{\boldsymbol{a}} = \underset{\boldsymbol{a}}{\arg\max}\, P_{\text{LLM}}(\boldsymbol{a}|c, \hat{\mathbb{X}}, p^{(sa)}) \tag{7}$$

Where $\hat{\boldsymbol{a}}$ represents the estimated assessment scores for each of the 10 symptom dimensions. The detailed prompt is provided in Appendix B

**Remark: Avoiding Excessive Speculation.** ClientCAST (Wang et al., 2024a), which simulates the client's estimation of psychometric test scores, our approach avoids excessive speculation. By adjusting the range of psychometric test scores to account for items not yet addressed by the client, our method more accurately reflects the gradual disclosure of information over time or across multiple sessions, preventing incomplete or biased assessments due to initially unmentioned items.

### 3.3 SESSION-FOCUSED CLIENT-DYNAMICS ASSESSMENT

Given the assessment scores $\hat{\boldsymbol{a}}$ of client information $c$, we use them to calculate Positive Symptom Distress Index (PSDI) (Derogatis & Unger, 2010), which quantifies the level of distress associated with positive psychological symptoms. The PSDI is calculated by averaging the distress scores assigned to each symptom, providing a numerical indication of the severity and impact of these

symptoms on the client's overall mental health. The PSDI is mathematically expressed by the formula:

$$\text{PSDI} = \frac{1}{N} \sum_{i \in \mathbb{P}} \hat{a}_i \tag{8}$$

Where $N$ is the number of positive symptoms, and $\hat{a}_i$ is the distress score for the $i$-th symptom, and $\mathbb{P}$ is the set containing the indices of all positive symptoms.

Consider a client whose initial stage information is denoted as $c_i$ and final stage information after completing treatment as $c_f$. By applying Equation 5 and 7 to the client information at each stage, we can obtain the initial stage assessment scores $\hat{a}_i$ and final stage assessment scores $\hat{a}_f$. Further, we can calculate the PSDI for both the initial and post-treatment stages using Equation 8 to assess the impact of treatment on the client's positive psychological symptoms. We define the change in symptoms as

$$e := \Delta\text{PSDI} = \text{PSDI}_f - \text{PSDI}_i \tag{9}$$

Where $\Delta\text{PSDI}$, defined as treatment outcome evaluation $e$ for the session $s$ in this work, represents the change in the PSDI from before to after treatment, quantifying the impact of the intervention on the client's distress related to positive psychological symptoms.

**Remark: Advantages and Versatility of PSDI.** Although PSDI is originally derived from the SCL-90, the method of calculating the average score of positive items offers the advantage of focusing directly on the relevant items of a psychometric test, leading to a more precise evaluation of treatment outcomes. This approach is not limited to the SCL-90 and can be easily extended to other psychometric tests, providing a flexible and reliable tool for assessing progress across different stages of treatment.

### 3.4 THERAPHASE DATASET

Popular datasets such as High-Low Quality (Pérez-Rosas et al., 2019), and AnnoMI (Wu et al., 2023), which contain client information primarily in the form of a single session, originate from public video sharing sources. These datasets only include client information relevant to the current stage and do not provide data for subsequent stages. To assess the changes in clients across different stages, we have constructed the TheraPhase Dataset based on the CPsyCoun (Zhang et al., 2024), which exhibits significant changes during a single session. Our dataset includes 400 pairs of client information from both the initial and completion stages of treatment.

**Construction Process.** To construct the TheraPhase Dataset, we utilize a 5-shot prompting approach with GPT-4 to extract the initial stage information from a client's comprehensive information. This method isolates the beginning portion of the client's data, forming a paired dataset where each pair consists of the initial client information and the corresponding full client information. This setup allows for an analytical comparison between the initial conditions and the outcomes after therapeutic interventions. The statistics of the resulting dataset are listed in Table 3.

## 4 EXPERIMENTS

In this section, we first conduct a psychological assessment based on various LLMs, evaluating their capability to detect and assess symptoms. Subsequently, we investigate their performance in predicting treatment outcomes.

### 4.1 EXPERIMENTAL SETTINGS

**IPAEval Setting Up.** The IPAEval framework is capable of handling various forms of client information, such as user profiles and interaction histories. However, due to data acquisition limitations, we primarily utilized consultation dialogue data as the main source of client information. Furthermore, IPAEval supports a variety of symptom-based psychometric tests, such as the General Health Questionnaire (GHQ) series (Montazeri et al., 2003), the Symptom Checklist (SCL) series, and the Brief Symptom Inventory (BSI) (Derogatis & Melisaratos, 1983). In this experiment, we

utilized the Symptom Checklist-90 (SCL-90) (Derogatis et al., 1973), a widely recognized and comprehensive tool for assessment a broad range of psychological symptoms. The scoring criteria for assessing symptoms, as set up and outlined in Table 2. Additionally, to ensure structured output, our code utilizes LangChain[1] and Pydantic[2] for better LLMs integration and data validation.

**Datasets.** We have selected two datasets for psychological assessment, High-Low Quality Counseling (Pérez-Rosas et al., 2019) and An-noMI (Wu et al., 2023), which consist of counseling therapy transcripts extracted from publicly available videos on online platforms such as YouTube and Vimeo. However, there are issues of data duplication between these two datasets. Given the higher quality of data in An-

Table 2: Scoring Criteria for Symptom Assessment

| Score | Description |
| --- | --- |
| -1 | Symptom not addressed. |
| 0 | Symptom addressed, but no symptoms found; no signs of distress or dysfunction. |
| 1 | Minimal symptoms, minor indications of distress but no significant dysfunction. |
| 2 | Clear symptoms, clear indications of distress, and significant dysfunction. |

noMI, we have chosen to retain the AnnoMI data from the same sources. Furthermore, considering the context window limitation of one of our test models, GPT-4, the maximum number of dialogue turns is set to 102. To increase the testing challenge and ensure the dialogues are sufficiently complex for evaluating the model's capability in handling extended therapeutic conversations, the minimum number of turns is set at 25. Based on these criteria, we have selected 110 client dialogue entries as our test data.

For treatment outcomes, we have selected the TheraPhase Dataset. This dataset comprises treament session transcripts that encompass two distinct phases of client interactions. Its advantage lies in the clear changes observable in clients across these phases, which aids in observing the treatment outcomes. The statistics of the resulting datasets are listed in Table 3.

| Datasets | Language | # of Clients | # of Sessions | Avg. # of Utterances | Words per Utterance |
| --- | --- | --- | --- | --- | --- |
| High-Low Quality Counseling AnnoMI | English | 110 | 110 | 79.8 (std = 26.1) | 22.2 (std = 27.1) |
| TheraPhase | Chinese | 400 | 800 | 11.5 (std = 6.3) | 41.7 (std = 20.9) |

Table 3: Summary of key characteristics of the selected datasets, including language, number of clients, sessions, average number of utterances per session, and the average word count per utterance.

**Evaluation Metrics.** We conducted a psychological assessment of LLMs focusing on two main aspects, symptom detection and symptom severity assessment. For symptom detection, we evaluated the model's ability to identify symptoms from a broad range of client information using classification metrics such as Accuracy, Precision, Recall, and F1 scores (Binary, Macro, and Weighted), based on scoring criteria from Table 2 where -1 indicates a negative class and 0, 1, and 2 represent positive classes. For assessing symptom severity, we calculated the PSDI score for each client and used error metrics such as Mean Squared Error (MSE) and Mean Absolute Error (MAE). To gauge the model's reliability, we reported the mean and standard deviation of these evaluation metrics across three runs, providing insight into the model's consistency in performance.

In evaluating treatment outcomes, we focus on the change in positive symptom severity, represented by $\Delta$PSDI, which reflects the difference in mean positive symptom scores between two assessments. A $\Delta$PSDI greater than 0 indicates a worsening of symptoms or the emergence of new ones, while a value less than or equal to 0 suggests symptom maintenance or improvement. We further evaluated the accuracy of predicting the direction of treatment outcome changes using metrics such as Accuracy, Precision, Recall, and F1 scores (Binary, Macro, and Weighted).

**References Generation.** To evaluate the performance of the models on psychological assessment tasks, we first required a set of reference scores for symptom detection and severity assessment. However, due to the lack of existing labeled data, we manually annotated 30 randomly selected client sessions. This manual annotation was carried out by two co-authors of this paper, both with

---

[1] https://www.langchain.com/
[2] https://docs.pydantic.dev/

significant expertise in natural language processing (NLP) and mental health research. The annotation process achieved a Cohen's kappa coefficient of 0.73, indicating substantial agreement between annotators. Following the annotation, we tested the performance of four closed-source models: GPT-4, GPT-4o, GPT-4-turbo, and GPT-4o-mini. The results, as shown in Table 4, indicated that GPT-4o outperformed the other models in both symptom detection and severity assessment. Based on these findings, GPT-4o was selected as the Gold Model for generating reference scores in psychological assessment tasks.

A similar issue arose in the treatment outcomes evaluation task. To address this, we followed the same approach as in the psychological assessment task. We manually annotated 60 sessions corresponding to 30 clients, focusing on their treatment outcomes. This annotation was again conducted by the two co-authors, achieving a Cohen's kappa coefficient of 0.81, reflecting a high level of agreement. The results, as presented in Table 5 shows that GPT-4 achieved the highest performance, thus it was chosen as the Gold Model for generating reference scores in treatment outcomes evaluation task.

| Models | Accuracy ↑ | Precision ↑ | Recall ↑ | $F1_{Binary}$ ↑ | $F1_{Macro}$ ↑ | $F1_{Weighted}$ ↑ | MSE ↓ | MAE ↓ |
|---|---|---|---|---|---|---|---|---|
| GPT-4 | $0.7744_{\pm0.01}$ | $0.6792_{\pm0.01}$ | $0.7187_{\pm0.01}$ | $0.6984_{\pm0.01}$ | $0.7591_{\pm0.01}$ | $0.7757_{\pm0.01}$ | $0.1369_{\pm0.02}$ | $0.2398_{\pm0.01}$ |
| GPT-4o | $0.7833_{\pm0.02}$ | $0.6674_{\pm0.02}$ | $0.8043_{\pm0.03}$ | $0.7295_{\pm0.02}$ | $0.7744_{\pm0.02}$ | $0.7867_{\pm0.02}$ | $0.1207_{\pm0.01}$ | $0.2272_{\pm0.02}$ |
| GPT-4-turbo | $0.7800_{\pm0.01}$ | $0.7503_{\pm0.03}$ | $0.5933_{\pm0.01}$ | $0.6623_{\pm0.01}$ | $0.7495_{\pm0.01}$ | $0.7734_{\pm0.01}$ | $0.2379_{\pm0.03}$ | $0.3754_{\pm0.03}$ |
| GPT-4o-mini | $0.4844_{\pm0.04}$ | $0.4079_{\pm0.02}$ | $0.9144_{\pm0.04}$ | $0.5634_{\pm0.01}$ | $0.4641_{\pm0.05}$ | $0.4370_{\pm0.06}$ | $0.1962_{\pm0.03}$ | $0.3265_{\pm0.02}$ |

Table 4: Comparison of different models on various performance metrics using human-annotated data in psychological assessment. Metrics with an upward arrow ↑ indicate higher values are better, while metrics with a downward arrow ↓ indicate lower values are better. The results show mean values along with standard deviations for each metric. Cells highlighted in blue represent the best-performing results.

| Models | Accuracy ↑ | Precision ↑ | Recall ↑ | $F1_{Binary}$ ↑ | $F1_{Macro}$ ↑ | $F1_{Weighted}$ ↑ |
|---|---|---|---|---|---|---|
| GPT-4 | $0.7444_{\pm0.06}$ | $0.8285_{\pm0.04}$ | $0.8406_{\pm0.04}$ | $0.8344_{\pm0.04}$ | $0.6370_{\pm0.08}$ | $0.7423_{\pm0.06}$ |
| GPT-4o | $0.6778_{\pm0.06}$ | $0.8219_{\pm0.02}$ | $0.7391_{\pm0.07}$ | $0.7770_{\pm0.05}$ | $0.5939_{\pm0.05}$ | $0.6916_{\pm0.05}$ |
| GPT-4-turbo | $0.6778_{\pm0.06}$ | $0.8046_{\pm0.02}$ | $0.7681_{\pm0.11}$ | $0.7815_{\pm0.05}$ | $0.5660_{\pm0.04}$ | $0.6809_{\pm0.04}$ |
| GPT-4o-mini | $0.7222_{\pm0.04}$ | $0.8625_{\pm0.06}$ | $0.7681_{\pm0.05}$ | $0.8090_{\pm0.03}$ | $0.6410_{\pm0.08}$ | $0.7306_{\pm0.05}$ |

Table 5: Comparison of different models on various performance metrics using human-annotated data in treatment outcomes.

**Models.** We conducted an investigation into the performance of several closed-source and open-source LLMs. The closed-source models we tested include GPT-4 (OpenAI et al., 2024), GPT4o, GPT-4-turbo, and GPT-4o-mini, which represent the latest advancements in proprietary LLMs developed by OpenAI[3]. . Additionally, we tested a variety of open-source models, such as Llama3.1-405B (Dubey et al., 2024), Llama3.1-70B (Dubey et al., 2024), Qwen2-72B (Yang et al., 2024), Mistral-8X22B (Jiang et al., 2024), and Mistral-8X7B (Jiang et al., 2024). These models vary significantly in terms of architecture, parameter size, and training data, providing a comprehensive overview of both commercial and community-driven LLM development. All of these models were invoked through API platforms[4]

## 4.2 MAIN EVALUATION RESULTS

**Psychological Assessments.** As shown in Table 6, GPT-4 achieved the best performance in symptom detection, excelling in both accuracy and binary F1 score, highlighting its strong ability to accurately identify symptoms. GPT-4-turbo demonstrated a more conservative approach with higher precision but lower recall, indicating it was more cautious in detecting symptoms but missed more cases. GPT-4o-mini excelled in recall but had reduced overall reliability due to a higher rate of false positives. Among open-source models, Qwen2-72B and Llama3.1-70B showed the closest performance to GPT-4, though they still fell short. Notably, Mistral-8X7B's extremely low recall was caused by a significant number of output formatting errors, leading to evaluation failures. We will further discuss these formatting issues in Appendix D.

---

[3]Specific versions of the OpenAI models used in the tests were `gpt-4-0613`, `gpt-4o-2024-05-13`, `gpt-4-turbo-2024-04-09`, `gpt-4o-mini-2024-07-18`.

[4]For the OpenAI models, we invoked them via https://platform.openai.com, Mistral models through https://console.mistral.ai/, Llama3.1 models via https://fireworks.ai/, and Qwen2 through https://www.together.ai/.

In symptom severity assessment, GPT-4 once again stood out with the lowest MSE and MAE, making it the most accurate model. Although GPT-4o-mini and GPT-4-turbo showed more balanced results, they were less precise compared to GPT-4. Among open-source models, Llama3.1-70B performed the best, though the gap between open-source and closed-source models remained substantial. Furthermore, GPT-4 exhibited the greatest consistency and reliability, with minimal variance across runs, indicating robust performance. In contrast, GPT-4o-mini showed more variability in MAE and MSE, and open-source models generally exhibited less stability compared to their closed-source counterparts.

| Models | Accuracy ↑ | Precision ↑ | Recall ↑ | $F1_{Binary}$ ↑ | $F1_{Macro}$ ↑ | $F1_{Weighted}$ ↑ | MSE ↓ | MAE ↓ |
|---|---|---|---|---|---|---|---|---|
| *Closed-Source Models* | | | | | | | | |
| GPT-4 | $0.7973_{\pm0.01}$ | $0.7852_{\pm0.01}$ | $0.7121_{\pm0.01}$ | $0.7469_{\pm0.01}$ | $0.7889_{\pm0.01}$ | $0.7956_{\pm0.01}$ | $0.2100_{\pm0.02}$ | $0.3292_{\pm0.03}$ |
| GPT-4-turbo | $0.7561_{\pm0.00}$ | $0.8726_{\pm0.02}$ | $0.4913_{\pm0.01}$ | $0.6285_{\pm0.01}$ | $0.7234_{\pm0.00}$ | $0.7386_{\pm0.00}$ | $0.4055_{\pm0.05}$ | $0.4490_{\pm0.03}$ |
| GPT-4o-mini | $0.4915_{\pm0.00}$ | $0.4467_{\pm0.02}$ | $0.8824_{\pm0.01}$ | $0.5931_{\pm0.00}$ | $0.4576_{\pm0.01}$ | $0.4359_{\pm0.01}$ | $0.2245_{\pm0.01}$ | $0.3329_{\pm0.02}$ |
| *Open-Source Models* | | | | | | | | |
| Llama3.1-405B | $0.7291_{\pm0.00}$ | $0.6960_{\pm0.01}$ | $0.6306_{\pm0.00}$ | $0.6616_{\pm0.00}$ | $0.7179_{\pm0.00}$ | $0.7269_{\pm0.00}$ | $0.3922_{\pm0.03}$ | $0.4476_{\pm0.01}$ |
| Qwen2-72B | $0.7385_{\pm0.00}$ | $0.7405_{\pm0.01}$ | $0.5815_{\pm0.01}$ | $0.6513_{\pm0.01}$ | $0.7210_{\pm0.00}$ | $0.7322_{\pm0.00}$ | $0.3962_{\pm0.01}$ | $0.4559_{\pm0.00}$ |
| Llama3.1-70B | $0.7333_{\pm0.01}$ | $0.7201_{\pm0.01}$ | $0.5974_{\pm0.01}$ | $0.6529_{\pm0.01}$ | $0.7182_{\pm0.01}$ | $0.7286_{\pm0.01}$ | $0.3379_{\pm0.01}$ | $0.4041_{\pm0.01}$ |
| Mistral-8X22B | $0.6215_{\pm0.00}$ | $0.5405_{\pm0.00}$ | $0.6616_{\pm0.02}$ | $0.5948_{\pm0.00}$ | $0.6198_{\pm0.00}$ | $0.6238_{\pm0.00}$ | $0.5205_{\pm0.03}$ | $0.5452_{\pm0.02}$ |
| Mistral-8X7B | $0.6070_{\pm0.00}$ | $0.6158_{\pm0.01}$ | $0.1710_{\pm0.02}$ | $0.2672_{\pm0.02}$ | $0.4993_{\pm0.01}$ | $0.5364_{\pm0.01}$ | $1.5711_{\pm0.02}$ | $1.0927_{\pm0.01}$ |

Table 6: Performance comparison between closed-source and open-source models across various evaluation metrics in psychological assessment.

**Treatment Outcomes.** Table 7 compares the performance of closed-source and open-source models on treatment outcome evaluation tasks. Among the closed-source models, GPT-4-turbo achieved the highest scores across multiple metrics, making it the most effective model in treatment outcome prediction. GPT-4o and GPT-4o-mini displayed competitive performance but lagged slightly behind GPT-4-turbo. For the open-source models, Llama3.1-405B led the group with the highest accuracy and macro F1, demonstrating superior performance in treatment outcome tasks. Qwen2-72B and Llama3.1-70B also performed well, while Mistral-8X7B had the highest recall but struggled with lower F1 scores, indicating higher sensitivity but less consistent overall performance. Overall, both closed-source and open-source models showed strong capabilities, with GPT-4-turbo and Llama3.1-405B emerging as the top performers in their respective categories.

| Models | Accuracy ↑ | Precision ↑ | Recall ↑ | $F1_{Binary}$ ↑ | $F1_{Macro}$ ↑ | $F1_{Weighted}$ ↑ |
|---|---|---|---|---|---|---|
| *Closed-Source Models* | | | | | | |
| GPT-4o | $0.6375_{\pm0.02}$ | $0.7706_{\pm0.01}$ | $0.7356_{\pm0.02}$ | $0.7526_{\pm0.01}$ | $0.5370_{\pm0.02}$ | $0.6448_{\pm0.02}$ |
| GPT-4-turbo | $0.6800_{\pm0.01}$ | $0.7824_{\pm0.01}$ | $0.7944_{\pm0.01}$ | $0.7883_{\pm0.00}$ | $0.5660_{\pm0.01}$ | $0.6772_{\pm0.01}$ |
| GPT-4o-mini | $0.6317_{\pm0.01}$ | $0.7727_{\pm0.01}$ | $0.7211_{\pm0.01}$ | $0.7459_{\pm0.01}$ | $0.5380_{\pm0.01}$ | $0.6420_{\pm0.01}$ |
| *Open-Source Models* | | | | | | |
| Llama3.1-405B | $0.6958_{\pm0.01}$ | $0.7965_{\pm0.01}$ | $0.7989_{\pm0.01}$ | $0.7976_{\pm0.00}$ | $0.5925_{\pm0.02}$ | $0.6951_{\pm0.01}$ |
| Qwen2-72B | $0.6725_{\pm0.01}$ | $0.7747_{\pm0.01}$ | $0.7944_{\pm0.01}$ | $0.7844_{\pm0.01}$ | $0.5515_{\pm0.01}$ | $0.6679_{\pm0.01}$ |
| Llama3.1-70B | $0.6708_{\pm0.01}$ | $0.7796_{\pm0.01}$ | $0.7822_{\pm0.01}$ | $0.7809_{\pm0.01}$ | $0.5597_{\pm0.02}$ | $0.6703_{\pm0.01}$ |
| Mistral-8X22B | $0.6383_{\pm0.01}$ | $0.7544_{\pm0.00}$ | $0.7678_{\pm0.01}$ | $0.7610_{\pm0.01}$ | $0.5089_{\pm0.01}$ | $0.6350_{\pm0.01}$ |
| Mistral-8X7B | $0.6825_{\pm0.01}$ | $0.7469_{\pm0.00}$ | $0.8722_{\pm0.02}$ | $0.8046_{\pm0.01}$ | $0.4779_{\pm0.00}$ | $0.6413_{\pm0.00}$ |

Table 7: Performance comparison between closed-source and open-source models across various evaluation metrics in treatment outcomes.

**Impact of Parameters on Performance.** As shown in Figure 3, model parameter size has a clear impact on performance across tasks such as symptom detection, symptom severity evaluation, and treatment outcome prediction. Larger models consistently outperform smaller models, exhibiting higher F1 (Weighted) scores and lower MAE. This trend indicates that increasing model size enhances the model's ability to handle complex tasks (Wen et al., 2024), especially in identifying subtle patterns related to psychological symptoms and predicting treatment outcomes.

### 4.3 ABLATION STUDY

**Impact on Items-aware Reasoning.** The ablation study, as shown in Figure 4, demonstrates the significant impact of items-aware reasoning on both psychological assessment and treatment outcomes

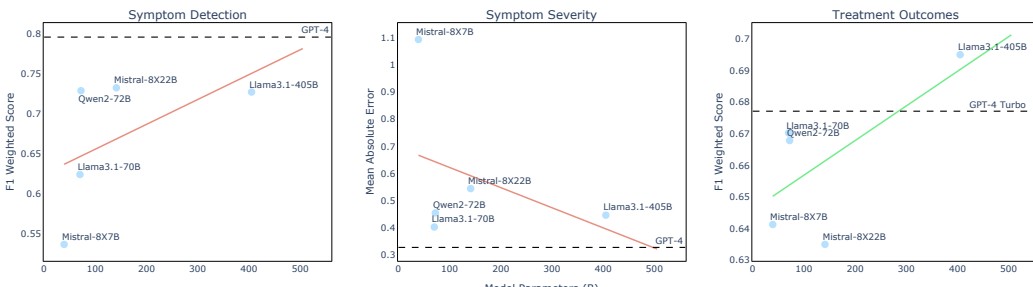

Figure 3: The impact of model parameters on symptom detection, symptom severity evaluation, and treatment outcome prediction. Dashed lines represent the best-performing closed-source models.

evaluation tasks. Removing this feature led to a substantial decline in performance across all models. For psychological assessment tasks, models like GPT-4o and GPT-4 experienced noticeable drops in their ability to accurately detect symptoms and assess severity, as reflected by decreases in F1 scores and increases in error metrics. Similarly, in treatment outcomes evaluation, the absence of items-aware reasoning resulted in reduced performance, though the impact was less pronounced compared to psychological assessment. These results underscore the importance of items-aware reasoning in improving the precision of the models in these tasks.

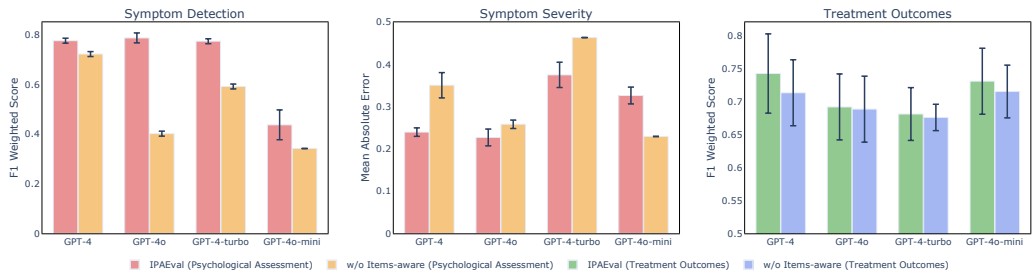

Figure 4: The impact of items-aware reasoning on psychological assessment and treatment outcomes evaluation using human-annotated data across four OpenAI models.

## 5  CONCLUSION

We introduced **IPAEval**, which can address the limitations of existing therapeutic outcome evaluation approaches by shifting the focus from therapist-centered, single-session assessments to a comprehensive, client-informed framework. By leveraging clinical interviews and integrating both cross-session client-contextual and session-focused client-dynamics assessments, IPAEval provides a more holistic evaluation of treatment outcomes. Experiments on the TheraPhase dataset validate its effectiveness in tracking symptom severity and therapeutic progress over multiple sessions, demonstrating significant improvements over previous single-session models. This advancement highlights the importance of client-centered, multi-session evaluations for enhancing mental health care and guiding treatment adjustments.

## LIMITATIONS

The limitations of this paper are as follows: (1) Due to the shortage of professional psychological annotators, only two individuals were involved in a limited amount of data labeling. This resulted in fewer human-aligned experimental data. Future research should focus on developing more multi-session datasets that include psychological assessment scores. (2) As the amount of client information increases, smaller models with fewer parameters struggle to follow instructions effectively. This limits the scalability and performance of these models in more complex scenarios. Future research should explore strategies to enhance model adaptability in handling larger client information inputs.

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

## A  ITEMS-AWARE REASONING PROMPTS IN EXPERIMENTS

---

**Prompt: Items-Aware Reasoning**

**Role:**
Imagine you are a skilled psychologist adept at identifying potential symptoms from interview. You can explain how these symptoms relate to the SCL-90 symptom checklist and specific items within it.

**Directives:**
Your task is to determine the presence or absence of symptoms from the Client's statements and provide detailed reasons for your assessment. Extract specific parts related to SCL-90 symptoms from the Client's statements. For each extracted part, indicate whether the symptom is present or not, and explain why this text is related to the SCL-90 symptom and specific item. If a symptom is mentioned but not present, extract that part and explain why the symptom is not present.SCL-90 is a psychological symptom assessment tool with 90 items, each evaluating different aspects of psychological distress.

**Additional Information:**
Symptom Checklist-90:
<Psychometric Test>
Presence of Symptoms: Extract the relevant part of the Client's statement. Indicate that the symptom is present. Explain why this text indicates the presence of the SCL-90 symptom and specify the item. Absence of Symptoms: Extract the part where the symptom is mentioned but not present. Indicate that the symptom is not present. Explain why this text does not indicate the presence of the SCL-90 symptom despite the mention.

**Output Formatting:**
<Format Instructions>

**Client Information:**
<Interview>

Please extract specific parts related to SCL-90 symptoms from the Client's statements. For each extracted part, indicate whether the symptom is present or not, and explain why this text is related to the SCL-90 symptom and specific item. If a symptom is mentioned but not present, extract that part and explain why the symptom is not present.

---

## B PSYCHOLOGICAL ASSESSMENT PROMPTS IN EXPERIMENTS

---

**Prompt: Psychological Assessment**

**Role:**
As a psychologist specializing in this evaluation task, based on the following interview and the extracted Symptom Checklist-90 (SCL-90) symptom-related content and explanation, provide a qualitative score (-1-2) for each symptom category.

**Score Criteria:**
Scoring criteria: -1 (Symptom not addressed in the interview), 0 (Symptom addressed in the interview, but no symptoms found, no signs of distress or dysfunction), 1 (Minimal symptoms, minor indications of distress but no significant dysfunction), 2 (Clear symptoms, clear indications of distress and significant dysfunction).

**Directives:**
Please note that this qualitative assessment is based on the state at the end of the interview. There may be noticeable symptoms during the interview, but these symptoms may become clarified or alleviated as the discussion progresses.

**Additional Information:**
<Psychometric Test>

**Output Formatting:**
<Format Instructions>

**Client Information:**
<Interview>
<Item-aware Reasoning Result>

Please extract specific parts related to SCL-90 symptoms from the Client's statements. For each extracted part, indicate whether the symptom is present or not, and explain why this text is related to the SCL-90 symptom and specific item. If a symptom is mentioned but not present, extract that part and explain why the symptom is not present.

---

SESSION:
Therapist: So, thank you for coming in today.
Client: Yes.
Therapist: How are you feeling today?
Client: I feel great actually.
Therapist: Yeah? Good.
Client: Yeah.
Therapist: Good.
Client: I feel good.
Therapist: And so you did your clarifications, value clarifications-
Client: Yeah.
Therapist: -and what are your top five?
Client: Yes. It was a good, uh, experience for me. It was different. It was different than usual. There were several things that were different, and, uh, the number one value that I put was self-respect. And I-I don't even know if self-respect has ever been in my top five let alone my number one.
Therapist: Really?
Client: Yeah. And, um—
Therapist: Do you have any idea why that is?
Client: I do have an idea, I think, why that is. Um, I think that there's been a few things that have happened recently and something that really came to my awareness, when I visited with my family, is that **I have consistently through my whole life, probably, put other people first. And I have consistently, uh, almost not even considered myself in the equation.** It was, uh, kind of sad in a way, at the time that I realized it. Uh, I didn't realize how severe it actually was, but I was kind of glad that I realized it because I feel like it's never too late to change-
Therapist: True.
Client: -and I feel like I can- I can, uh, respect and value myself just as much as I have other people. I know that's important. And I feel like when I do that, I'm a better person for other people as well.
Therapist: Mm-hmm. By not putting yourself on the back burner so much?
......

ITEMS-AWARE REASONING RESULT:
Client Statement: **I have consistently through my whole life, probably, put other people first. And I have consistently, uh, almost not even considered myself in the equation.**
Symptom Category: Interpersonal Sensibility
Specific Symptom: Feeling others do not understand the client or are unsympathetic.
Presence: Yes
Explanation: The client's statement indicates that they have been prioritizing others over themselves, which could be a sign of feeling misunderstood or not receiving empathy from others.
......

ASSESSMENT SCORE:
.......; Interpersonal Sensitivity: 1;......

Table 8: Items-Aware Reasoning Output Example

## C    ITEMS-AWARE REASONING OUTPUT EXAMPLE

## D    OUTPUT FORMATTING ERRORS

In our two experiments, OpenAI series models produced no errors in output formatting, whereas open-source models encountered numerous issues. Specifically, the Figure 5 below shows the error statistics for open-source models during the Assessment task, with the main issue being incorrect output that did not follow the Pydantic-defined JSON format.

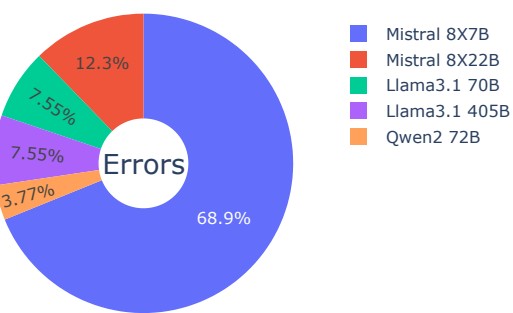

Figure 5: Error distribution across different models.

