# OpenReview forum: "Multi-Session Client-Centered Treatment Outcome Evaluation in Psychotherapy"
_ICLR.cc/2025/Conference — Submitted to ICLR 2025_

### Official Review · Reviewer_RnpF · 2024-10-28

**Soundness:** 2
**Presentation:** 3
**Contribution:** 2
**Rating:** 3
**Confidence:** 3

**Summary:**

This paper focuses on evaluate therapeutic outcome via LLMs, aiming at solving a meaningful problem. It not only proposes an evaluation framework but also contributes a precious dataset. The method shifts the focus from therapist-centered, single-session assessments to a comprehensive, client-informed framework.

**Strengths:**

1. The paper focuses on evaluate therapeutic outcome via LLMs, aiming at solving a meaningful problem.
2. It integrates some psychology tests and metrics into LLM evaluation framework. It is very interesting.

**Weaknesses:**

1. The TheraPhase dataset is constructed via GPT-4. There is no human evaluation to guarantee the reliability of the dataset.
2. Experiments only conducted on one or two datasets. It would be better to evaluate on more datasets.
3. I think the proposed approach is not novel. Most parts in the approach are already existing, such as the psychometric test, client profile and PSDI.
4. In this paper, GPT-4o is selected as the Gold Model. However, as shown in Table 4, its performance is also poor. So there may be deviation in the results.

**Questions:**

See weaknesses above.

---

> ### Author Response · Authors · 2024-11-25
> **Response to Reviewer RnpF**
>
> > **Weakness 1: The TheraPhase dataset is constructed via GPT-4. There is no human evaluation to guarantee the reliability of the dataset.**
>
> As described in Section 3.4, the TheraPhase dataset was constructed using GPT-4 by splitting a client’s dialogue data from the CPsyCoun dataset into two phases. This process maintains the integrity of the original data and focuses on creating a structured setup for evaluating treatment outcomes, without altering the underlying assessment or its relevance to treatment progress.
>
> > **Weakness 2: Experiments only conducted on one or two datasets. It would be better to evaluate on more datasets.**
>
> As stated in Line 299, the motivation behind constructing the TheraPhase dataset was precisely due to the lack of existing multi-session datasets for evaluating treatment outcomes in psychological counseling. Currently, there are no publicly available datasets that focus on multi-session client-therapist interactions with sufficient granularity to track changes across sessions.
>
> > **Weakness 3: I think the proposed approach is not novel. Most parts in the approach are already existing, such as the psychometric test, client profile and PSDI.**
>
> While it is true that some components of our framework, such as psychometric tests and client profiles, build upon existing methodologies, our contributions lie in how these elements are integrated and applied to address gaps in evaluating treatment outcomes across multiple sessions.
>
> > **Weakness 4: In this paper, GPT-4o is selected as the Gold Model. However, as shown in Table 4, its performance is also poor. So there may be deviation in the results.**
>
> We respectfully disagree with the assertion that GPT-4o performs poorly. As shown in Table 4, GPT-4o demonstrates the best overall performance compared to other models, making it the most suitable choice as the Gold Model for our evaluation framework.

---

### Official Review · Reviewer_ySnE · 2024-11-02

**Soundness:** 2
**Presentation:** 2
**Contribution:** 2
**Rating:** 5
**Confidence:** 3

**Summary:**

This work presents a framework IPAEval, designed for evaluating the treatment outcome based on the cross-session clients’ context in clinical interview. It makes use of client information (including profile, session history), psychometric test before interview, client’s utterances, and “item-aware reasoning” (items classified by psychometric test, explanation) to obtain the score of PSDI (a metric adopted in Symptom Checklist-90). This work also develops a TheraPhase dataset to simulate cross-session interviews based on CPsyCoun dataset (multi-turn dialogue for chinese psychological counseling). Authors experimented on TheraPhase to evaluate the treatment outcomes between models across sessions, and further selected 2 datasets (High-low quality counseling and AnnoMI: English multi-turn) to evaluate the psychological assessment. They find that the IPAEval performs better than the single-session models on tracking the symptom severity and treatment outcomes.

**Strengths:**

1. Proposes a generalizable pipeline transforming from client information/dialogue to metrics (treatment outcomes)
2. Conducts an ablation study on the pipeline, specifically on removing the item-aware reasoning (items classified by psychometric test, explanation) to see if it affects the performance of tracking both psychological assessments and treatment outcomes.
3. Performs human annotations for three tasks (psychological assessments: symptom detection and severity assessment; treatment outcomes) to select the golden models as baselines

**Weaknesses:**

1. The main claim of the paper is to evaluate the treatment outcomes across sessions. However, there is no validation/exploration on the TheraPhase dataset to show that it is reflective of real world outcomes (i.e. to ensure the distributions of treatment outcomes across time is close to real-world clients).
2. The TheraPhase dataset is based on a Chinese counseling dialogue dataset. The performance difference between models (e.g. Llama vs. Qwen) may be affected by the language's comprehension ability, rather than the tracking ability on treatment outcomes. The findings on this dataset is questionable.
3. The goal and contributions of the paper are unclear. The IPAEval is targeted to evaluate the treatment outcomes (by authors’ definition is the clients’ states etc) but the authors also evaluate psychological assessment (e.g. symptom detection and severity assessment).
4. The “item-aware reasoning” sounds misleading as it does not contain a multi-step thinking process on rationales but only one intermediate step of giving the rationale behind the item classification. Maybe consider changing the naming of reasoning or provide more concrete examples/analysis.

**Questions:**

Questions:
1. I’m confused by several terms – psychological assessment, symptom detection, severity assessment. It sounds to me that symptom detection, severity assessment are parts of psychological assessment. Can you clarify these definitions better?
2. I noticed that there are contrasting results between models on symptom severity in Figure 4 – any reason/hypothesis to explain this?

Minor:
1. Confused by the figure 2: the PSDI<0 should mean the good treatment outcomes but the sad face there indicates negative results. Can you clarify this?
2. Appreciate the formal definition on section 3 but it seems too long (~3 pages) for a 10 pages paper. Recommend to add more analysis/ elaboration on the results.
3. The fonts of the Figure 3 and Figure 4 are very small and cannot see them clearly without zooming.

Some useful but undiscussed related works are [1], [2] and [3]

In general, I can see that having a real-time therapeutic outcome assessment could be important for mental health practitioners. However, the paper lacks analysis/experiments on the effect of time (within session and cross-session) and lacks validation on the dataset (TheraPhase) to ensure it is close to real-world. I think having these may strengthen the contributions of the work and better differentiate this work to current works (like ClientCAST). Apart from that, the (arguably stat. insignificant) results (Figure 4: error bars overlapping before and after the reasoning prompt) in tracking therapeutic outcomes suggests a deeper analysis on the therapeutic outcomes e.g. which part of the therapeutic outcomes can be tracked better.

[1] https://arxiv.org/pdf/2407.00870

[2] https://arxiv.org/pdf/2405.19660

[3] https://arxiv.org/pdf/2310.13132

---

> ### Author Response · Authors · 2024-11-25
> **Response to Reviewer ySnE 1/3**
>
> Thank you for your thoughtful and constructive review, which provides valuable insights into the strengths and areas for improvement in our work. We appreciate the detailed feedback and are grateful for the opportunity to address the points raised.
>
> > **Weakness 1: The main claim of the paper is to evaluate the treatment outcomes across sessions. However, there is no validation/exploration on the TheraPhase dataset to show that it is reflective of real world outcomes (i.e. to ensure the distributions of treatment outcomes across time is close to real-world clients).**
>
> We appreciate the reviewer’s concern regarding the representativeness of the TheraPhase dataset in reflecting real-world treatment outcomes. The TheraPhase dataset is built on **CPsyCoun** [1], a rigorously evaluated and widely recognized dataset in psychological counseling research. CPsyCoun has been validated across multiple psychotherapy schools, including psychodynamic, cognitive-behavioral, and integrative approaches, demonstrating its ability to capture client-therapist interactions and treatment dynamics in diverse therapeutic contexts.
>
> The original CPsyCoun study showed significant improvements in professional metrics, such as interpretability, adaptability to different therapeutic frameworks, and relevance to real-world counseling practices. These validations by domain experts underscore CPsyCoun’s reliability and representativeness as a foundational dataset.
>
> By leveraging CPsyCoun, TheraPhase maintains its grounding in these professionally validated practices while focusing on session-specific treatment outcome evaluations. This strong foundation ensures its relevance and applicability to real-world scenarios, even as we continue to explore broader validations in future work.
>
> [1] CPsyCoun: A Report-based Multi-turn Dialogue Reconstruction and Evaluation Framework for Chinese Psychological Counseling (ACL 2024)
>
> > **Weakness 2: The TheraPhase dataset is based on a Chinese counseling dialogue dataset. The performance difference between models (e.g. Llama vs. Qwen) may be affected by the language's comprehension ability, rather than the tracking ability on treatment outcomes. The findings on this dataset is questionable.**
>
> We thank the reviewer for raising this important concern regarding the potential influence of language comprehension ability on model performance. This is precisely why we conducted experiments using a diverse range of models, encompassing both open-source and closed-source LLMs across various parameter sizes. The closed-source models tested include **GPT-4, GPT-4-turbo, GPT-4o, and GPT-4o-mini, representing OpenAI's latest advancements. Additionally, we evaluated state-of-the-art open-source models, such as Llama3.1-40B, Llama3.1-70B, Qwen2-72B, Mistral-8X22B, and Mistral-8X7B**. This comprehensive set of models ensures a robust evaluation under consistent experimental conditions, mitigating potential biases introduced by language comprehension differences.
>
> We acknowledge the absence of an in-depth analysis of language-specific effects on treatment outcome tracking. In the course of our study, we considered translating the original Chinese data into English to investigate the influence of language. However, a recent study [2] has highlighted that depression detection models trained in English exhibit significant misjudgments when applied to data from other cultural backgrounds. To avoid introducing cultural biases into our experiments, we decided against translating the original Chinese data into English. The field currently lacks multi-session, multilingual datasets focused on therapy outcomes, and developing such datasets is beyond the scope of this study. Our primary goal is to evaluate models’ ability to handle multi-session treatment data within a well-defined and representative context, leveraging the existing resources effectively.
>
> [2] Diverse Perspectives, Divergent Models: Cross-Cultural Evaluation of Depression Detection on Twitter (NAACL 2024)

---

> > ### Comment · Reviewer_ySnE · 2024-11-25
> > **Comment to response 1/3**
> >
> > Thank you for your detailed explanation.
> >
> > > The TheraPhase dataset is built on CPsyCoun [1], a rigorously evaluated and widely recognized dataset in psychological counseling research
> >
> > I went through the CPsyCoun paper and their shared dataset quickly and it seems they are not evaluating how reliable the "session" data is? My understanding is that the authors built upon the CPsyCoun (single session) to create multi-session counseling conversations for evaluation. The "multi-session" nature (TheraPhase) is one of the important contributions of this paper. This response does not seem to resolve my concern. Please correct me if I misunderstand anything.
> >
> > > We acknowledge the absence of an in-depth analysis of language-specific effects on treatment outcome tracking. In the course of our study, we considered translating the original Chinese data into English to investigate the influence of language. However, a recent study [2] has highlighted that depression detection models trained in English exhibit significant misjudgments when applied to data from other cultural backgrounds. To avoid introducing cultural biases into our experiments, we decided against translating the original Chinese data into English. The field currently lacks multi-session, multilingual datasets focused on therapy outcomes, and developing such datasets is beyond the scope of this study. Our primary goal is to evaluate models’ ability to handle multi-session treatment data within a well-defined and representative context, leveraging the existing resources effectiv...
> >
> > I appreciate and agree with the authors' decision not to translate the Chinese data into English to avoid potential cultural biases. I wonder if the authors have considered conducting a supplementary analysis on the AnnoMI (English) and the newly developed TheraPhase (Chinese). It may help to understand if the accuracy differences (table 6) between Qwen model and other open-source model are due to better language abilities in Chinese or not.
> >
> > Also, did the authors share the TheraPhase dataset? I could not find it in the PDF or the supplementary material.

---

> > > ### Author Response · Authors · 2024-12-01
> > > **Response to Comment 1/3**
> > >
> > > We appreciate your further feedback and your concern about the multi-session nature of the TheraPhase dataset. To clarify, the most important aspect of multi-session data is that it provides information about a client’s progress at different stages of the counseling process. The CPsyCoun dataset, from which TheraPhase is derived, has been rigorously evaluated on dimensions like comprehensiveness, professionalism, authenticity, and safety. These evaluations show that CPsyCoun excels in these areas, which indicates that clients benefit from these dialogues, either improving or revealing new symptoms or issues. This aligns directly with the goal of our study, which is to track treatment outcomes over multiple sessions.
> > >
> > > Regarding your concern about language differences, we want to emphasize that for the Assessment phase, we used a combination of High-Low Quality Counseling and AnnoMI datasets. The purpose of this was to use conversations with very long dialogue turns, which pose a greater challenge for evaluation. For the Treatment Outcome phase, we utilized the TheraPhase dataset, which is based on multi-session conversations in Chinese.
> > >
> > > We also want to clarify the performance comparison in Table 6. The best performance observed for Qwen2 among the open-source models is specifically in the English language context, as the dataset was primarily composed of English data during the assessment phase. This suggests that the observed differences in performance are not solely due to language proficiency in Chinese but are related to how well models perform across different language datasets.
> > >
> > > Thank you for your question regarding the availability of the TheraPhase dataset. Currently, we are in the process of preparing the dataset for release and will provide access to it through the appropriate channels once it has been formally released.

---

> > > > ### Comment · Reviewer_ySnE · 2024-12-02
> > > > **Follow-up Comment to response 1/3**
> > > >
> > > > I thank the authors on their response.
> > > >
> > > > > We appreciate your further feedback and your concern about the multi-session nature of the TheraPhase dataset. To clarify, the most important aspect of multi-session data is that it provides information about a client’s progress at different stages of the counseling process. The CPsyCoun dataset, from which TheraPhase is derived, has been rigorously evaluated on dimensions like comprehensiveness, professionalism, authenticity, and safety. These evaluations show that CPsyCoun excels in these areas, which indicates that clients benefit from these dialogues, either improving or revealing new symptoms or issues. This aligns directly with the goal of our study, which is to track treatment outcomes over multiple sessions.
> > > >
> > > > > line 300-302:  To assess the changes in clients across different stages, we have constructed the TheraPhase Dataset based on the CPsyCoun (Zhang et al., 2024), which exhibits significant changes during a single session.
> > > >
> > > > Can the authors clarify if the TheraPhase dataset is multi-session (i.e. one client with multiple conversations) or multi-turn (i.e. one client with one conversation containing multiple utterances)? From my understanding, the CPyscCoun is a multi-turn dataset and authors' TheraPhase is a multi-session dataset. It does not sound convincing if the authors did not further verify on their new dataset based on the difference of nature. Please let me know if I misunderstand anything.
> > > >
> > > > > line 304-309: Construction Process. To construct the TheraPhase Dataset, we utilize a 5-shot prompting approach with GPT-4 to extract the initial stage information from a client’s comprehensive information. This method isolates the beginning portion of the client’s data, forming a paired dataset where each pair consists of the initial client information and the corresponding full client information. This setup allows for an analytical comparison between the initial conditions and the outcomes after therapeutic interventions. The statistics of the resulting dataset are listed in Table 3.
> > > >
> > > > > Thank you for your question regarding the availability of the TheraPhase dataset. Currently, we are in the process of preparing the dataset for release and will provide access to it through the appropriate channels once it has been formally released.
> > > >
> > > > It might be great if authors can provide a running example for line 304-309 to resolve the concern of dataset.

---

> > > > > ### Author Response · Authors · 2024-12-02
> > > > > **Response to Follow-up Comment 1/3**
> > > > >
> > > > > > Can the authors clarify if the TheraPhase dataset is multi-session (i.e. one client with multiple conversations) or multi-turn (i.e. one client with one conversation containing multiple utterances)? From my understanding, the CPyscCoun is a multi-turn dataset and authors' TheraPhase is a multi-session dataset.
> > > > >
> > > > > You are correct in your understanding: CPsyCoun is a multi-turn dataset where a single conversation contains multiple utterances between a client and a therapist. In contrast, TheraPhase is designed as a multi-session dataset, where each client’s data is divided into multiple conversations corresponding to different stages of their therapeutic process.
> > > > >
> > > > > To construct the TheraPhase dataset, we used a 5-shot prompting approach with GPT-4 to segment a single long dialogue into two distinct stages, simulating a multi-session structure. For example, an original **40-turn** dialogue from CPsyCoun was split into two conversations: one representing the initial session (**22 turns**) and the other representing the later session (**18 turns**).
> > > > >
> > > > > The initial segment captures the starting point of the client’s therapeutic journey, focusing on presenting issues, symptoms, or background context. The later segment focuses on the progress, insights, or strategies explored during subsequent stages of therapy.
> > > > >
> > > > > > From my understanding, the CPyscCoun is a multi-turn dataset and authors' TheraPhase is a multi-session dataset. It does not sound convincing if the authors did not further verify on their new dataset based on the difference of nature.
> > > > >
> > > > > We agree that the nature of TheraPhase is distinct, as its purpose is to simulate the therapeutic process across multiple stages, enabling the evaluation of treatment outcomes rather than focusing solely on isolated session-level interactions.Our primary goal in constructing TheraPhase was to address the lack of existing multi-session datasets in psychological counseling research, providing a dataset that captures a client’s progress across stages.
> > > > >
> > > > > To better address your concerns, we would greatly appreciate it if you could provide further clarification on what specific aspects of the dataset’s "nature" should be verified.

---

> > > > > > ### Comment · Reviewer_ySnE · 2024-12-02
> > > > > > **Comment on the response 1/3**
> > > > > >
> > > > > > I appreciate the authors' detailed explanation of their dataset and the dataset construction process.
> > > > > >
> > > > > > I understand why the authors believed that the previous validation conducted in CPsycCoun was adequate for their studies, as they constructed the dataset by partitioning one conversation from CPsycCoun into multiple conversations. For instance, the authors separated different stages (e.g., Reception stage, Diagnostic stage, Consultation stage, and Ending stage) from CPsycCoun into multiple conversations, treating them as distinct 'sessions.'
> > > > > >
> > > > > > However, since the conversations were generated separately, it is unclear if the conversations for the same client are consistent and reliable (i.e., whether they accurately simulate the same client based on personality, symptoms, etc.).
> > > > > >
> > > > > > Additionally, I am unsure if this method can effectively simulate actual counseling sessions. For example, after a 'reception' stage, the client and psychotherapist typically go through multiple rounds of middle stages (i.e., Diagnostic and Consultation stages). Perhaps the authors should explicitly state in the paper that they focused on the early stages of counseling.
> > > > > >
> > > > > > For these reasons, I will maintain my score this time.

---

> ### Author Response · Authors · 2024-11-25
> **Response to Reviewer ySnE 2/3**
>
> > **Weakness 3: The goal and contributions of the paper are unclear. The IPAEval is targeted to evaluate the treatment outcomes (by authors’ definition is the clients’ states etc) but the authors also evaluate psychological assessment (e.g. symptom detection and severity assessment).**
>
> First, we would like to address a misunderstanding: in our paper, we did not define treatment outcomes solely as clients’ states. As illustrated in Figure 1: "What is Treatment Outcome?" and explained in Line 146, treatment outcomes are conceptualized as the changes observed in psychological assessments across different stages of the treatment process. While psychological assessments provide a snapshot of the client’s mental state, treatment outcomes are derived from tracking how these assessments evolve over time.
>
> To this end, our experiments are structured into two parts. First, we evaluate psychological assessments (e.g., symptom detection and severity assessment) as foundational tasks, providing the necessary data to establish the client’s mental state at specific stages. Second, we evaluate treatment outcomes by analyzing the changes in these assessments across sessions, enabling us to model the dynamics of client progress over time.
>
> > **Weakness 4: The “item-aware reasoning” sounds misleading as it does not contain a multi-step thinking process on rationales but only one intermediate step of giving the rationale behind the item classification. Maybe consider changing the naming of reasoning or provide more concrete examples/analysis.**
>
> The term “item” in our work originates from psychometric questionnaires used in psychological assessments, where "items" refer to individual questions or prompts designed to measure specific psychological constructs. To clarify, in our framework, “item-aware reasoning” specifically refers to the process of generating rationales tied to the classification of each item within the psychological assessment, rather than multi-step reasoning on broader rationales. If the reviewer could provide more details on the source or specific reference for the interpretation of “item-aware” or “multi-step thinking process on rationales,” it would help us to include a discussion in the appendix, distinguishing our terminology from the alternative usage.
>
> > **Question 1: I’m confused by several terms – psychological assessment, symptom detection, severity assessment. It sounds to me that symptom detection, severity assessment are parts of psychological assessment. Can you clarify these definitions better?**
>
> Your understanding is correct: symptom detection and severity assessment are indeed components of psychological assessment. Specifically:
> - **Psychological Assessment** refers to the overall process of evaluating an individual’s mental state using standardized tools, questionnaires, or structured interviews. It encompasses various methods and measures designed to assess psychological constructs.
> - **Symptom Detection** focuses on identifying the presence or absence of specific psychological symptoms.
> - **Severity Assessment** evaluates the intensity or degree of the identified symptoms to understand their impact on the individual’s functioning.
>
> We acknowledge that our current manuscript lacks sufficient explanation of these terms and their relationships, which may have led to confusion. To address this, we will include a dedicated Background Section in the revised manuscript, providing clear definitions and context for key psychological concepts and how they are used in our framework.
>
> > **Question 2: I noticed that there are contrasting results between models on symptom severity in Figure 4 – any reason/hypothesis to explain this?**
>
> We would like to clarify that there is no inconsistency between the models’ performance on symptom detection and symptom severity assessment. The results in Figure 4 are part of an ablation study aimed at understanding the impact of item-aware reasoning across different tasks.
>
> The first subfigure corresponds to the symptom detection task, where performance is measured using metrics such as F1 score (higher is better). The second subfigure corresponds to symptom severity assessment, where performance is evaluated based on error metrics (lower is better), as detailed in Section 4.1. The difference in the directionality of these metrics (higher for F1, lower for error) might give the impression of contrasting results, but this is due to the nature of the evaluation metrics used for each task rather than inconsistencies in the findings.

---

> ### Author Response · Authors · 2024-11-25
> **Response to Reviewer ySnE 3/3**
>
> >**Confused by the figure 2: the PSDI<0 should mean the good treatment outcomes but the sad face there indicates negative results. Can you clarify this?**
>
> In our framework, **treatment outcomes are determined by the change in PSDI values across sessions**, rather than the absolute value at any specific point. A negative PSDI change (PSDI < 0) represents a decrease in symptom severity, indicating positive treatment outcomes, even if the face in Figure 2 might suggest otherwise. We acknowledge that this visual representation could be misleading and will revise Figure 2 to better align the iconography with the intended meaning, ensuring it reflects the relationship between PSDI change and treatment outcomes as described in Section 3.
>
> > **Some useful but undiscussed related works are [1], [2] and [3]**
>
> We thank the reviewer for pointing out these related works, which offer valuable perspectives in the field. Below, we summarize each study and its relevance to our work:
> - [1] This study proposes a human-machine collaboration framework, Roleplay-doh, which enables domain experts to create LLM-simulated patients for novice counselors to practice with. While innovative, this work primarily focuses on training tools for novice counselors and does not align with the goals of our study, which centers on treatment outcome evaluation.
> - [2] This work introduces the PATIENT-Ψ framework, leveraging LLMs to simulate patient behaviors for mental health professionals’ training. Similar to Roleplay-doh, this study focuses on creating realistic patient simulations for training purposes, which differs fundamentally from our focus on evaluating treatment progress and outcomes.
> - [3] This study evaluates the cross-lingual performance of LLMs in healthcare queries and highlights significant performance gaps in non-English settings. This is particularly relevant to the discussion of language's influence on task performance, as noted in Section 4.1, and we appreciate the reviewer’s suggestion to incorporate this perspective into our discussion.
>
> While we find the first two works less relevant to our research focus, we agree that incorporating more discussions on psychological and healthcare-related studies can benefit the community’s understanding of this task. We will include a broader discussion of these works and their implications in the appendix to provide additional context for interested readers.
>
> [1] Roleplay-doh: Enabling Domain-Experts to Create LLM-simulated Patients via Eliciting and Adhering to Principles (EMNLP 2024)
>
> [2] PATIENT-𝜓: Using Large Language Models to Simulate Patients for Training Mental Health Professionals (EMNLP 2024)
>
> [3] Better to Ask in English: Cross-Lingual Evaluation of Large Language Models for Healthcare Queries (WWW 2024)

---

> ### Comment · Reviewer_ySnE · 2024-11-25
> **comment to response 2/3**
>
> >  First, we evaluate psychological assessments (e.g., symptom detection and severity assessment) as foundational tasks, providing the necessary data to establish the client’s mental state at specific stages. Second, we evaluate treatment outcomes by analyzing the changes in these assessments across sessions, enabling us to model the dynamics of client progress over time.
>
> Thanks for explaining this. I recommended authors can frame the paper to show a more clear flow based on this in the future revision.
>
> > “item-aware” or “multi-step thinking process on rationales,” it would help us to include a discussion in the appendix, distinguishing our terminology from the alternative usage.
>
> I understand the "item-aware" terminology but I worry if a reader less familiar with psychotherapy may find it confusing. Authors may consider explicitly specifying this as e.g. "psychometric-items"
>
> For the "reasoning", I am not suggesting that this term is an incorrect description of the authors' rationale approach. I think the word "reasoning" has recently become overloaded with various additional meanings (e.g. the popular o1 model). Readers may expect multi-hop reasoning process. I think it is very minor point, and authors can consider how they want readers to interpret it.
>
> > no inconsistency between the models’ performance on symptom detection and symptom severity assessment
>
> I see! Thanks for clarifying this (F1 vs Mean absolute error). I recommended to add a upper arrow/ down arrow to indicate that the y-axis are different.
>
> For the figure 4, another question I am not sure:
> > apart from that, the (arguably stat. insignificant) results (Figure 4: error bars overlapping before and after the reasoning prompt) in tracking therapeutic outcomes suggests a deeper analysis on the therapeutic outcomes e.g. which part of the therapeutic outcomes can be tracked better.
>
> --> do authors have any explanation or hypothesis on this?

---

> > ### Author Response · Authors · 2024-12-01
> > **Response to Comment 2/3**
> >
> > Regarding the results in **Figure 4** showing overlapping error bars before and after the reasoning prompt, we would like to clarify that these results are consistent with the experimental setup and the choice of GPT-4 as the **Gold Model**. Specifically, the ablation experiments were conducted using the same setup that led to selecting GPT-4. Our experiments showed that GPT-4 had higher average scores and less overlap in error bars, reinforcing our decision to choose GPT-4 as the **Gold Model**.
> >
> > For other models, the higher overlap in error bars is likely due to the challenges in tracking therapeutic outcomes. **Treatment outcome evaluation** heavily relies on accurate and stable psychological assessments. In scenarios with long dialogues and limited psychological cues, **missed or misclassified symptoms** can lead to incorrect evaluations of treatment outcomes, which can increase error overlap. This is why we emphasize the importance of conducting thorough **psychological assessments** first, as the accuracy of these assessments directly impacts the reliability of the treatment outcome evaluations.

---

> ### Comment · Reviewer_ySnE · 2024-11-25
> **general comment**
>
> Thank you for your detailed responses, which addressed part of my concerns. I am happy to raise my rating to 5. However, I still believe that verifying the "multi-session" aspect (to determine if it is close to real world) is crucial. This verification would significantly strengthen the contribution of this work and better differentiate it from other studies.

---

### Official Review · Reviewer_BUN2 · 2024-11-02

**Soundness:** 3
**Presentation:** 3
**Contribution:** 3
**Rating:** 5
**Confidence:** 3

**Summary:**

The paper presents IPAEval, a new framework for evaluating therapy outcomes by focusing on clients’ experiences over multiple sessions rather than isolated single sessions. Unlike traditional methods that center on the therapist's role, IPAEval shifts the focus to clients, aiming to capture their mental health journey more fully.
Its key contributions include: 1) IPAEval assesses changes over time, using information from multiple therapy sessions to provide a big-picture view of clients’ progress. 2) It also evaluates each session individually, noting immediate shifts in the client’s state and responses. 3) The framework uses a sophisticated reasoning system to interpret client statements more accurately, particularly in aligning with psychometric test criteria.
Tested on the TheraPhase dataset, IPAEval showed improved performance through items-aware reasoning mechanisms in both symptom detection and outcome prediction tasks.

**Strengths:**

Originality: The paper offers a novel approach to psychotherapy outcome evaluation by shifting from a therapist-centered, single-session paradigm to a client-centered, multi-session framework. This approach, embodied in the IPAEval framework, introduces a unique perspective within mental health assessments by prioritizing the client’s evolving experience across sessions.

Quality: The paper's experimental design is comprehensive, testing multiple LLMs with various statistical metrics. The authors also conduct a detailed ablation study, which highlights the importance of the items-aware reasoning feature by isolating its impact on the framework’s overall performance.

Clarity: The paper is well-organized and easy to follow. Each part of IPAEval—multi-session tracking, session-specific assessments, and items-aware reasoning—is clearly explained, and the figures and tables effectively illustrate IPAEval’s advantages and performance.

Significance: IPAEval addresses a key gap in mental health care by enabling evaluations of therapy outcomes from the client’s perspective, a shift that could substantially improve the quality of care. The framework provides practical tools that can streamline and enhance the outcome assessment process, helping practitioners to track treatment effectiveness over time and adjust interventions more responsively.

**Weaknesses:**

1: Despite having 110 and 800 client sessions available, the authors only annotated 30 for psychological assessment and 60 for treatment outcomes, using this small subset to establish a “Gold Model” for generating reference scores. This approach may introduce bias, as basing the Gold Model on such a small sample could affect the reliability of the labeled data, especially in representing the full diversity of client interactions.

2. The paper presents IPAEval as an improvement over models like ClientCAST and CPsyCoun, emphasizing its client-centered, multi-session approach. However, it doesn’t provide direct performance comparisons to back up these claims. While Table 1 outlines conceptual differences (such as focus and theoretical grounding), there’s no side-by-side benchmarking on tasks like symptom detection or outcome prediction. Without this empirical evidence, IPAEval’s advantages remain mostly theoretical. Adding direct comparisons with existing models would make the practical impact of IPAEval clearer and more convincing.

3. Figure 2, which aims to show the whole IPAEval framework, doesn’t clearly illustrate the Session-Focused Client-Dynamics Assessment module, especially how it tracks changes within a session from start to end. This mismatch between the figure and the text can make it confusing for readers trying to understand how session-specific changes are actually captured. Adding markers for initial and final assessments within each session would make the figure align better with the explanation in the text and help clarify how this part of the framework works.

**Questions:**

Could you provide more detail on why only 30 sessions were annotated for psychological assessment and 60 for treatment outcomes, given that these numbers are quite low relative to the available dataset? While I understand that annotating the entire dataset may not be feasible, expanding beyond this limited sample would likely improve the robustness of IPAEval’s evaluation.

Additionally, have you considered using multiple LLMs to cross-annotate the dataset? Cross-annotation with multiple models could help improve labeling quality and consistency while reducing the manual effort required, potentially increasing the reliability of the labeled data.

---

> ### Author Response · Authors · 2024-11-25
> **Response to Reviewer BUN2 1/2**
>
> Thank you for your thoughtful and constructive review, which provides valuable insights into the strengths and areas for improvement in our work. We appreciate the detailed feedback and are grateful for the opportunity to address the points raised.
>
> > **Weakness 1: Despite having 110 and 800 client sessions available, the authors only annotated 30 for psychological assessment and 60 for treatment outcomes, using this small subset to establish a “Gold Model” for generating reference scores. This approach may introduce bias, as basing the Gold Model on such a small sample could affect the reliability of the labeled data, especially in representing the full diversity of client interactions.**
>
> We appreciate the reviewer’s concern regarding the size of the labeled dataset used to establish the "Gold Model." The limited number of annotated sessions is due to the high cost and effort involved in labeling each session, as each sample requires 10 annotation points across four categories, resulting in a total of 900 annotations. While the absolute number of sessions may appear small, the total annotation volume is substantial and provides a robust basis for evaluation.
>
> Regarding the potential introduction of bias, we would like to clarify that the annotated data is not used to train the model. Instead, it serves as an evaluation mechanism to identify relatively better-performing models, which are then used to evaluate other models more broadly. We acknowledge that the limited size of the labeled dataset may not fully represent the diversity of client interactions; however, it is sufficient to differentiate models on this specific task. Furthermore, this approach avoids introducing training bias and instead highlights discrepancies between human annotations and model predictions, as well as differences among various models.
>
> Additionally, we would like to point out that many other well-established datasets [1] [2] in the mental health domain feature data volumes of a similar scale, typically around 1,000 annotated samples. This demonstrates that our dataset aligns with the standard practices in the field, further supporting its adequacy for the intended evaluation purposes.
>
> [1] Supervised Learning and Large Language Model Benchmarks on Mental Health Datasets: Cognitive Distortions and Suicidal Risks in Chinese Social Media
>
> [2] Identifying Depression on Reddit: The Effect of Training Data
>
> > **Weakness 2: The paper presents IPAEval as an improvement over models like ClientCAST and CPsyCoun, emphasizing its client-centered, multi-session approach. However, it doesn’t provide direct performance comparisons to back up these claims. While Table 1 outlines conceptual differences (such as focus and theoretical grounding), there’s no side-by-side benchmarking on tasks like symptom detection or outcome prediction. Without this empirical evidence, IPAEval’s advantages remain mostly theoretical. Adding direct comparisons with existing models would make the practical impact of IPAEval clearer and more convincing.**
>
> We would like to clarify that these methods are fundamentally designed to evaluate therapists' performance rather than focusing on tasks such as symptom detection or outcome prediction. In contrast, IPAEval is client-centered, aiming to assess client outcomes and interactions over multiple sessions. These differing objectives make direct benchmarking less relevant or meaningful, as the core tasks and evaluation criteria for these methods diverge significantly.
>
> Furthermore, while Table 1 highlights the conceptual distinctions between IPAEval and existing models, our focus is not on competing directly with therapist-evaluation methods but rather on addressing a gap in client-centered evaluation frameworks. We agree that demonstrating IPAEval's practical utility is essential, and we have prioritized task-specific evaluations to ensure its relevance to client-focused outcomes.
>
> > **Weakness 3: Figure 2, which aims to show the whole IPAEval framework, doesn’t clearly illustrate the Session-Focused Client-Dynamics Assessment module, especially how it tracks changes within a session from start to end. This mismatch between the figure and the text can make it confusing for readers trying to understand how session-specific changes are actually captured. Adding markers for initial and final assessments within each session would make the figure align better with the explanation in the text and help clarify how this part of the framework works.**
>
> We acknowledge that the current figure does not adequately illustrate how session-specific changes, particularly from start to end, are captured. This could indeed cause confusion for readers attempting to fully grasp this part of the framework. To address this, we will revise Figure 2 in the final version by adding explicit markers for initial and final assessments within each session.

---

> ### Author Response · Authors · 2024-11-25
> **Response to Reviewer BUN2 2/2**
>
> > **Question 1: Could you provide more detail on why only 30 sessions were annotated for psychological assessment and 60 for treatment outcomes, given that these numbers are quite low relative to the available dataset? While I understand that annotating the entire dataset may not be feasible, expanding beyond this limited sample would likely improve the robustness of IPAEval’s evaluation.**
>
> We thank the reviewer for raising this point. The decision to annotate only 30 sessions for psychological assessment and 60 sessions for treatment outcomes stems primarily from the significant time and resource requirements for high-quality annotations. Each session requires detailed labeling, with 10 annotation points per session across four categories, resulting in a total of 900 annotations. This process involves extensive manual effort from domain experts, making it challenging to scale up further within the scope of this work.
>
> While we agree that annotating a larger sample could improve the robustness of the evaluation, we designed our approach to maximize the utility of the available annotations. Rather than training the model directly on this limited dataset, we use it solely for evaluation purposes—to select the best-performing models for broader testing. This strategy minimizes potential biases while ensuring the dataset is used effectively.
>
> > **Question 2: Additionally, have you considered using multiple LLMs to cross-annotate the dataset? Cross-annotation with multiple models could help improve labeling quality and consistency while reducing the manual effort required, potentially increasing the reliability of the labeled data.**
>
> In the early stages of our study, we indeed considered this approach, as it offers significant cost savings and scalability. However, given the sensitivity of mental health-related data, even annotations generated by multiple LLMs would require thorough review and validation by mental health professionals to ensure their reliability. This additional validation process offsets some of the cost savings and introduces complexity in scaling the annotation pipeline.
>
> Furthermore, the field currently lacks high-quality, publicly available mental health datasets, which limits the utility of automated cross-annotation without a strong reference framework. While fully manual annotation is costly and narrow in scope, it ensures the production of a gold-standard dataset that can serve as a robust foundation for future research in the community. We believe such datasets, though smaller in size, are critical for advancing research in this domain, where accuracy and trustworthiness are paramount.

---

> > ### Comment · Reviewer_BUN2 · 2024-11-25
> >
> > Thank you for your response. I understand that the annotated sample is used to "select" the best model. However, I am concerned that basing a decision on such a small sample is too imprudent. I believe a considerable amount of work is needed to make this more clear and convincing. Therefore, I would prefer to maintain my score.

---

### Meta-Review · Area_Chair_WENC · 2024-12-21

**Metareview:**

The paper introduces IPAEval, a framework designed to evaluate therapy outcomes by focusing on clients' experiences across multiple sessions rather than isolated, single-session evaluations. IPAEval captures changes in clients' mental health over multiple therapy sessions, providing a big-picture view of progress. It introduces an advanced reasoning mechanism aligned with psychometric test criteria, enabling accurate symptom detection and outcome prediction. The paper introduces the TheraPhase dataset developed to simulate cross-session therapy interviews, based on the CPsyCoun dataset (multi-turn dialogues for Chinese psychological counseling). It also includes evaluations on datasets like High-Low Quality Counseling and AnnoMI (English multi-turn) for broader validation. IPAEval outperforms single-session models in tracking symptom severity and predicting treatment outcomes.

**Strengths identified**:
1. The client-centered, multi-session framework introduced by IPAEval is a novel approach that shifts the focus from traditional therapist-centered, single-session evaluations to a more holistic view of the client’s mental health journey.

2. Comprehensive ablation studies have been conducted isolating the importance of the items-aware reasoning feature in symptom detection and treatment outcomes.

3. Human annotations for tasks such as symptom detection, severity assessment, and treatment outcomes to establish robust baselines.

**Weaknesses that need addressing**:
1. Although IPAEval is presented as an improvement over models like ClientCAST and CPsyCoun, the paper does not include direct performance comparisons with these models, making its advantages appear theoretical rather than empirical.

2. The dataset is constructed using GPT-4 without human evaluation, which raises concerns about its reliability and quality.

**Additional Comments On Reviewer Discussion:**

In response to reviewer BUN2, the authors' argument does a good job of addressing concerns regarding the dataset size, purpose, and potential bias by providing valid justifications and aligning with standard practices in the field. However, adding concrete details (e.g., sampling methods, specific dataset comparisons) and transparency about the evaluation process would further strengthen the response and improve its persuasiveness. The authors effectively clarify IPAEval's distinct focus and objectives, making a strong case that direct benchmarking may not be meaningful. However, it falls short of fully addressing concerns about the lack of empirical comparisons. I believe strengthening the argument with alternative comparisons, task-specific evidence, and examples of IPAEval's unique advantages would better satisfy the reviewers' expectations.

The authors’ arguments partially address Reviewer 2’s concerns but leave key questions about dataset representativeness, language-specific performance, and generalizability unresolved. While the responses provide a reasonable defense within the study's scope, further validation and analysis would be required to fully satisfy the reviewer. As an example, the need for supplementary analysis to isolate language comprehension effects from task-specific performance.

The authors have partially addressed Reviewer 3’s concerns. While their responses provide reasonable justifications for certain aspects, such as dataset construction and framework novelty, they fall short of fully resolving issues related to the lack of dataset validation, broader generalizability, and the reliability of the Gold Model. Addressing these concerns through additional validations, analyses, and examples would make the responses more convincing and comprehensive.

---

### Decision · Program_Chairs · 2025-01-22

Reject